# Numerical modelling of loose top coal and roof mass movement for a re-mined seam using the top coal caving method

**Shuai Wang**[1]*, **Chunhua Zhang**[1], **Feng He**[2], **Yu Yang**[3]

**1** College of Safety Science and Engineering, Liaoning Technical University, Fuxin, China, **2** College of Mechanics and Engineering, Liaoning Technical University, Fuxin, China, **3** College of Civil Engineering, Liaoning Technical University, Fuxin, China

* zzbganbuke@l26.com

**Data Availability Statement:** All relevant data are within the paper.

**Funding:** This research was financially supported by National Natural Science Foundation of China (2017YFC1503102), and Liaoning Natural Science

## Abstract

The re-mined face using top coal caving system is the most applicable method for recovering the remaining reserves of a previous partially-mined thick coal seam. However, this mining method may encounter the problems of low recovery and unpredicted geological conditions. A numerical model is developed using PFC2D for studying the movement of top coal mass and development of coal-rock mass interface at a longwall top coal caving re-mined face. The re-mined face advances in the lower seam below the upper solid coal pillar, previous entries and gob. A theoretical analysis according to the unsteady flow model is developed to calculate the proper time duration of caving operation. The results showed that the top coal to be recovered through the caving window before the initiation of caving operation is a partial spheroid-shaped geometry. The boundary between the coal and rock mass then develops to a funnel-shaped coal-roof interface as the caving operation continues. The top coal recovery is 98.1%, 77.1% and 70.5% for caving operations below the solid coal, entries and gob area in the upper seam, respectively. The proper timing of caving and interval of caving operation is important for obtaining a high coal recovery. Good agreement is achieved between the proposed model and the improved Boundary-Release model for short of B-R model. The study in this work may provide reference for the safety and efficiency of the extraction of the longwall top coal caving re-mined face.

## 1 Introduction

Coal seam larger than 8 m is referred as the extreme thick seam, which is widely found in north and west part of China. The modern mining equipment and technology have now enabled exploiting a thick coal seam of 8–20 m at one single pass. The large-cutting-height longwall system has been successfully practiced in Shangwan mine, Inner Mongolia for mining an 8.8-m thick total seam with the most powerful longwall shearer and shields. For seams thicker than 10 m (mostly in Xinjiang and Shanxi), however, the top coal caving mining method remains the most effective and practicable method for extracting the full height seam

Foundation(Grant No.20180550869).The funders had no role in study design, data collection and analysis, decision to publish, or preparation of the manuscript.

at one pass. The longwall top coal caving method was first started in Europe in the 1950s and was then introduced in China in the 1980s. It soon gained its popularity nationwide, and has been successfully practiced in the industry for almost 40 years. The longwall top coal caving method is also applied to recover the re-mined thick coal seams, which refers to extracting the remaining reserves of a full thick coal seam that has been partially mined previously. This is because the thick seam was not fully extracted due to the geological complexity or limitation of mining technology. Therefore, the room and pillar system or slicing mining method was used for recovering part of the upper slicing for the time being, leaving both the remaining upper reserves and lower seam for future extraction.

Due to the improvement of mining equipment and the depletion of coal reserves in China, attention has been paid to recovering the remaining reserves of the previous partially mined seam. The top-coal caving mining method can be used for recovering the re-mined seam, since this mining method not only extracts the lower seam, but recovers the remaining reserves of the upper seam. However, the re-mined longwall face using top coal caving method is much more complicated and difficult than the conventional caving face. The upper seam of the conventional face is purely solid coal, while the re-mined face may encounter unpredicted geological conditions typically observed as unstable immediate roof (un-caved or partially-caved), cavities (entries or shortwall faces) and mined-out gob. Loose rock wastes, mine water and gas may be found in the cavity or gob. The re-mined seam using the top coal caving long-wall system includes the re-mined face on the lower seam advancing below previous entries (Fig 1A) and gob area in the upper seam (Fig 1B). The mining equipment is placed at the lower seam, with the front scraper conveyor ahead of shield for transporting the cutting coal and the rear scraper conveyor behind the shield for recovering the caving coal. The progressive development of the re-mined longwall face may advance through the solid coal, entries and gob area in the upper seam. The caving materials, therefore, may contain solid coal from the upper seam and rock wastes from gob or entries of previous upper faces.

Wang found that top coal failure and loose coal mass caving mechanisms remain the most important ground control problems for the top coal caving mining method [1]. The top coal recovery and waste ratio are two of the most meaningful parameters for assessment of the top coal failure and caving mechanisms. Note that the waste ratio is an instantaneous parameter that refers as the volume of waste rocks over the total volume of materials flowing through the caving window of longwall shield at a certain moment. This parameter is used for determining the proper timing for shutting down the caving window. Some other important influencing factors include the coal-roof mass boundary, caved coal geometry (the geometry of the amount of recovered top coal at initiation of caving operation), caving-cutting ratio (defined as the ratio of mining height to caving height), seam inclination, interval of caving operation, caving method and timing, etc. These parameters are commonly utilized for studying the top coal movement.

The field of longwall top coal caving has performed extensive studies by researchers [2–5]. Yu presented a structure evolution model with respect to strata behaviours, studied the behaviors of strata overlying the extra-thick coal seams with the combined method of theoretical analysis, physical simulation, and field measurement [6]. Zhang studied the development of coal-roof boundary and caving body geometry for different caving operations using physical and numerical modelling approaches [7]. Le presented a discontinuum modelling approach to investigate Longwall Top Coal Caving (LTCC) behaviour including stress distribution, coal and rock failures, top coal caving and roof strata rupture, and to analyse the impact of overburden movement on top coal caving [8]. Feng investigated the simultaneous recovery of upper remnant coal pillars while mining the ultra-close lower panel using longwall top coal caving (LTCC) using physical and numerical modelling approaches [9]. A BBR (B-Boundary of Top-

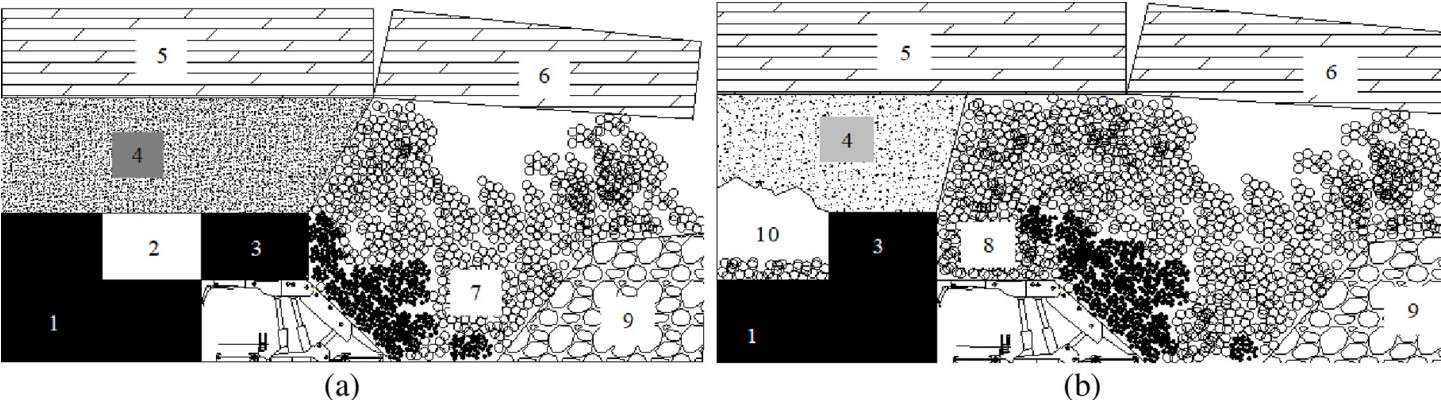

**Fig 1.** Re-mined face under (a) previous entries and (b) gob area in upper seam. 1. Lower seam;2. Previous entry; 3. Previous coal pillar; 4. Immediate roof; 5. Main roof; 6. Caved roof block; 7. Uncompacted gob materials from previous upper seam; 8. Caved materials sitting on the shield; 9. Compacted gob materials; 10. Previous unfilled gob in upper seam.

Coal,B-Drawing Body of Top-Coal,R-Recovery Ratio of Top-Coal and Rock Mixed Ratio of Top-Coal.) assessing system was developed for evaluating the parabolic-like coal-roof mass interface and consequently the caving geometry of coal body [10]. The B-R model (Boundary-Release model) based on mechanics of granular media was improved by modifying the gravitational acceleration factor, and was used for obtaining the best timing of caving operation at different seam thickness [11]. Wang improved the recovery ratio of the large interval caving operations for extra-thick seam based on the BBR theory [12]. Wang investigated the geometry of caving body and interface between the coal and the roof mass using the methods of 3D physical modeling, numerical modeling, and field observations [13]. Zhang presented an approach coupling the particle element and the block element based on the continuum discontinuum element method (CDEM), to investigate top-coal drawing regularity and automated top-coal caving technology in extra-thick coal seams [14]. The 3D distribution characteristics of caving coal and roof mass was analyzed at multiple partings in thick coal seams [15–19]. The influence of caving-cutting ratio and caving interval on caving geometry were included. Gong improved the caving parameters and top coal recovery by analysing the amount of lost coal [20]. Wang compared the influence of caving height on the displacement and the stress distribution of roof and floor using numerical modelling method [21]. Wu analysed the relationship between the critical size of caving window on the shield and the top coal particle dimensions [22].

These works have improved our understanding on mechanism of top coal failure and movement of loose top coal. However, the re-mined seam was not included in previous studies. The development of coal-rock interface requires further investigation for the re-mined seam using longwall top coal caving method. This paper attempts to extend the study of loose coal movement in the field of top coal caving re-mined face using the PFC 2D numerical modelling approach. An analytical analysis with the unsteady flow model of caving materials is also provided to analyse the flowing characteristics of the top coal mass and proper timing of cease of caving operation.

## 2 Model development

### 2.1 Mine description

The representative numerical simulation in this study is based on No. 2 coal seam at Xuehugou mine, Huozhou, Shanxi. The favourable geological condition of the coal mine is simple and no major geological disturbance is observed, though the coal seam contains well-developed coal cleats. The near-flat seam inclines toward the northwest at an angle of 1–7˚, at 80–170 m depth. The total thickness of the seam is 4–6 m (average at 4.92 m). The 4–6 m immediate roof above the coal seam is the weak and soft mudstone with well-defined bedding planes. The immediate roof caves in upon shield advance and breaks in small pieces of rock blocks. The 5–6 m main roof is fine-grained sandstone showing good stability. The immediate floor is a mixture of soft siltstone and sandy mudstone with a thickness of 3–5 m. No. 2 seam was originally worked randomly in an unplanned manner for partial extraction of the top 2.5–4.5 m slicing (average of 3 m). The recovery ratio is only 25–30%. The total reserves to be recovered from the lower seam remain over 5 Mt. The top coal caving method was applied to extract the remaining reserves in this coal mine. During the entry development of the re-mined face, it was found 50% of the entry roof were affected by the previous mining activities. Most of the cavities and gob areas in the upper seam are partially filled with the waste rocks from the weak immediate roof.

### 2.2 Model description

Fig 2 shows the developed PFC 2D numerical model with the partially mined-out upper seam. The model is 30 m long and 20 m high, with the wall elements as boundaries along the sides and at the bottom. The original velocity of the elements and particles is 0, and the acceleration of gravity is set at 9.80 m/s². According to the No. 2 seam at Xuehugou, the height of the lower and upper seam is designed as 3 m and 2 m, respectively, and the immediate roof is 4 m thick. The caving-cutting ratio is therefore 3:2. The top coal caving re-mined face is developing in the lower seam in 1.5 m increments from the opening cut at left side of the model to the right, advancing through the 3-m solid coal pillars, the 2-m wide entries and 3-m wide gob areas in the upper seam, so that the geological conditions can be covered (see below Fig 2). The caving operation is performed after 3 cuts of the longwall face, and is ceased when the waste ratio reaches 9–15%. The movement trajectories, interface of coal and rock mass, and the geometry of recovered top coal are recorded for assessing the movement of partially-mined top coal mass. Table 1 shows the rock mass properties used in modelling.

## 3 Results and discussion

### 3.1 Top coal movement

**3.1.1 Face advancing below solid coal pillar.** At the beginning of face development, the re-mined face advances below the solid coal pillar. Fig 3 shows the movement of top coal and roof rock mass after 4.5 m of face advance (corresponding to 5 cuts of the re-mined face). A large amount of top coal is sitting above the shield caving window before caving operation (Fig 3A). The V-shaped interface of the top coal and roof mass is observed. After caving, the majority of top coal flows through the caving window and is recovered by the rear conveyor (Fig 3B). As the coal-roof interface reaches the caving window, the V-shaped coal-roof boundary is significantly altered. The caving operation is ceased before large amount of waste rocks flowing through the caving window. The remaining coal mass in the gob area is the amount of lost coal that cannot be recovered. The particles close to the caving window show the largest velocity and displacement (Fig 3C and 3D).

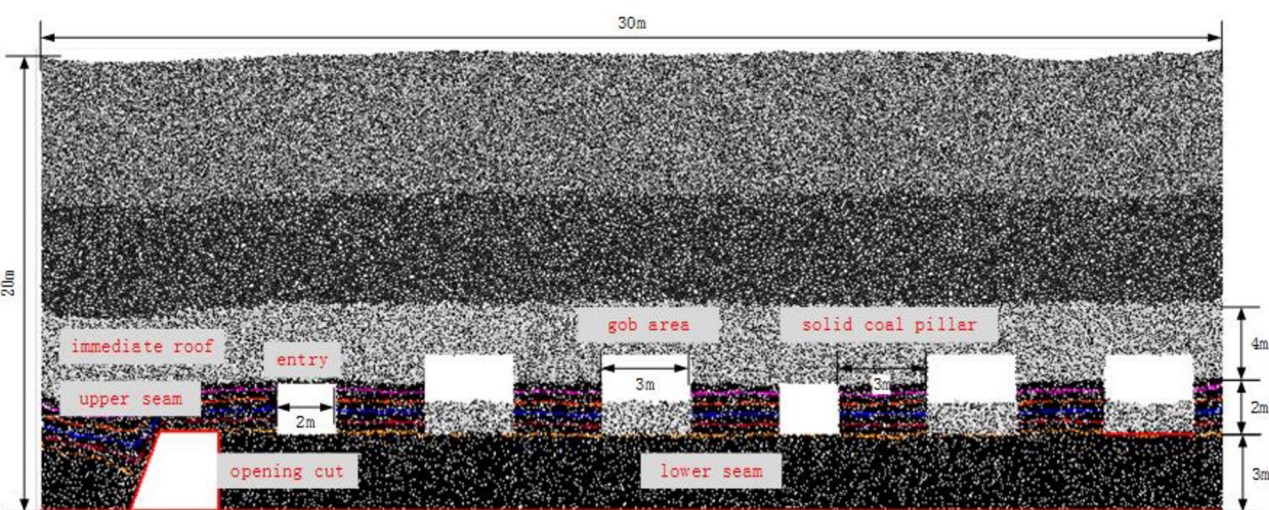

**Fig 2. Numerical model showing the layout of re-mined lower seam and the previously partially mined-out upper seam.**

**3.1.2 Face advancing below upper entry.** The re-mined face is moved to the area below the upper entries after 9 m of face advance. As compared with Fig 3A where caving is operated under solid coal, the amount of coal mass above the caving window is significantly reduced (see Fig 4A). Upon the caving operation, the top coal is recovered from the caving window at its own weight, and the coal-roof boundary is also significantly changed (Fig 4B). The caving operation ceases at the waste ratio of 7–12%. As compared to the face advancing under the solid coal pillar, less amount of un-recovered coal is left behind the shield and in the gob area. The velocity of particles is slightly higher at the caving window but lower at the lemniscate link, and the displacement of particles is also larger (Fig 4C and 4D).

**3.1.3 Face advancing below upper gob area.** The re-mined face reaches the upper gob area after 22.5 m of face advance, where the gob is partially filled with the immediate roof mass. The coal and roof caving characteristics and movement of the coal-roof mass boundary before and after caving are shown in Fig 5. The remaining top coal in upper seam is mainly resting above the caving window before the initiation of caving (Fig 5A). The top coal starts to flow through the caving window following the caving operation, and is recovered from the rear conveyor (Fig 5B). As the caving continues, however, the waste rocks above the remaining top coal show larger movement velocity than the remaining coal behind the shield (Fig 5C). Therefore wastes reach the caving window first, leaving part of the top coal in the gob area unrecovered. The waste ratio is about 10–15% after the cease of caving. The particles show the largest displacement (Fig 5D).

**Table 1. Rock mass properties used in numerical modelling.**

| Lithology | Normal stiffness, GPa | Shear stiffness, GPa | Internal friction angle, ° | Cohesion, MPa | Tensile strength, MPa | Particle radius, mm |
|---|---|---|---|---|---|---|
| Mudstone | 9 | 1.2 | 31 | 4 | 0.5 | 8~9 |
| Fine sandstone | 3 | 0.5 | 37 | 7 | 0.4 | 7~8 |
| Sandy mudstone | 3 | 0.5 | 30 | 2 | 0.1 | 3~4 |
| Coal | 2 | 0.5 | 25.2 | 2.49 | 0.1 | 3~6 |

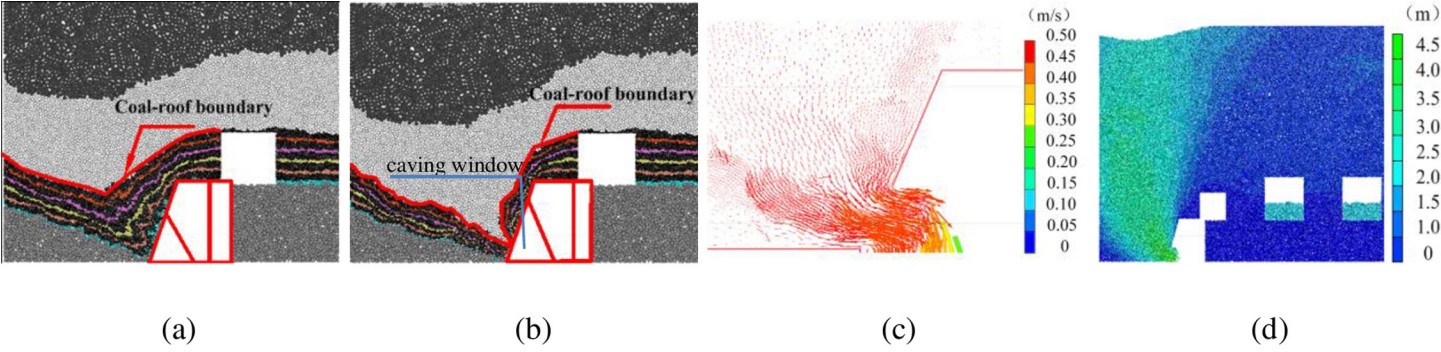

**Fig 3.** Coal and roof mass caving characteristics after 4.5 m of face advance below solid coal: (a) Coal-roof mass boundary before caving; (b) Coal-roof mass boundary after caving; (c) Particle velocity vectors at caving window; (d) Particle displacement.

### 3.2 Geometry of caved top coal body

Fig 6 shows the geometry of the recovered top coal mass above the caving window when the face advances below the upper entries and gob area. The figures are obtained by removing the recovered top coal mass from the before-caving geometry at the initiation of caving operation. A partial spheroid-shaped geometry of the caved top coal is observed (see the red solid lines in Fig 6). The axis of the spheroid is located close to the gob side rather on the shield window. The volume of caving coal mass on left side of the axis is slightly larger. The previously-caved immediate roof rocks in the upper entry and gob reduce the flowability of the top coal and move to the caving window prior to the remaining top coal. Therefore, unrecovered top coal is found behind the shield. More unrecovered coal mass remains behind the shield when the face advances below gob than the entry. The dimensions of the caving geometry below gob area are larger than that below the upper entry.

### 3.3 Movement of loose coal-roof mass interface

Fig 7 plots a schematic of the geometry of top coal above the caving window during caving operation and the progressive development of coal-roof boundary. The schematics are abstracts from the numerical modelling results in Fig 5. The top coal above the caving window is funnel-shaped after caving (see Fig 7A), which is developed from the partial spheroid-shaped geometry at initiation of caving (see Fig 6B). The progressive development of coal-roof boundary throughout the caving operation is given in Fig 7B. In this figure, the red circle represents a

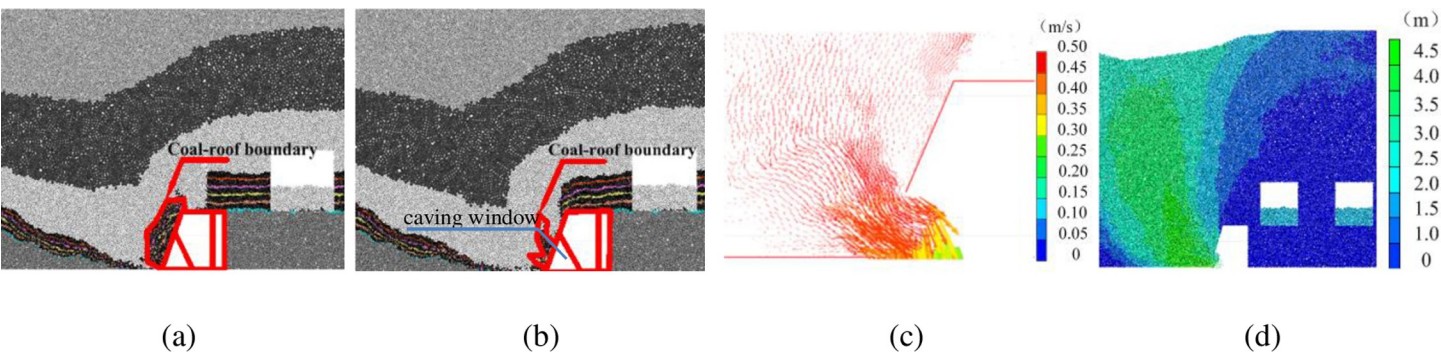

**Fig 4.** Coal and roof caving characteristics after 9m of face advance below previous entries in upper seam: (a) Coal-roof mass boundary before caving; (b) Coal-roof mass boundary after caving; (c) Particle velocity vectors at caving window; (d) Particle displacement.

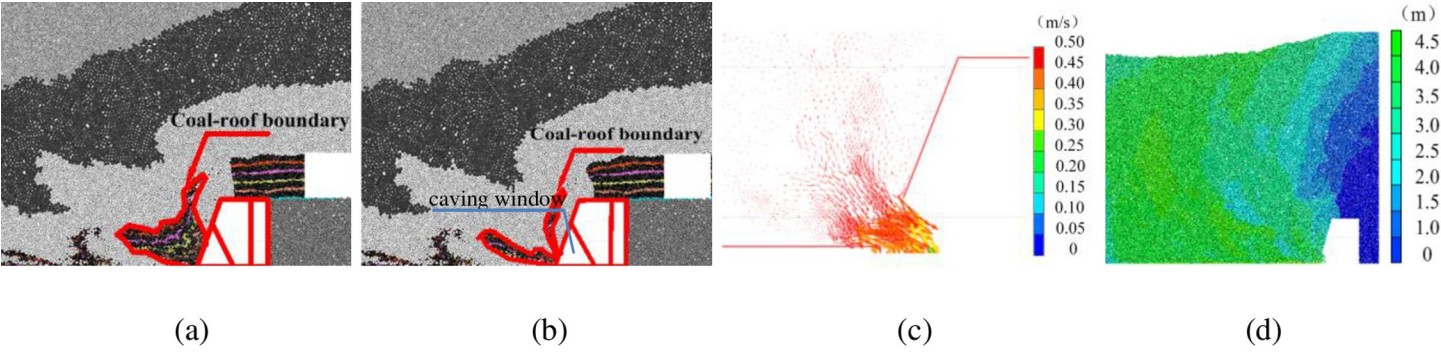

**Fig 5.** Coal and roof caving characteristics after 22.5m of face advance below gob areas in upper seam: (a) Coal-roof mass boundary before caving; (b) Coal-roof mass boundary after caving; (c) Particle velocity vectors at caving window; (d) Particle displacement.

mark point on the coal-roof boundary; the black arrow line represents the trajectory of the track point; and the shading represents the caving velocity of the coal body (warm colour represents a larger velocity). The discrepancy in movement velocity of the caving top coal is observed at different positions above the shield, and the coal mass close to the caving window shows the largest caving velocity.

## 3.4 Top coal recovery

The top coal recovery is calculated as the ratio of the recovered top coal over the sum of recovered and lost coal, and is shown in Table 2. The recovered and lost top coal is obtained by tracking the amount of coal recovered through the shield window and the amount left in the gob, respectively. It shows that the top coal recovery is 98.1% when the face advances below the solid coal, compared with 77.1% below entries and 70.5% below gob area.

The relationship between the waste ratio and top coal recovery is plotted in Fig 8. When the caving operation is performed below the solid coal, the materials coming through the caving window are coal mass at the beginning of caving operation. Rock wastes start to flow in the caving window when the top coal recovery reaches 50%. When the top coal recovery is 85%,

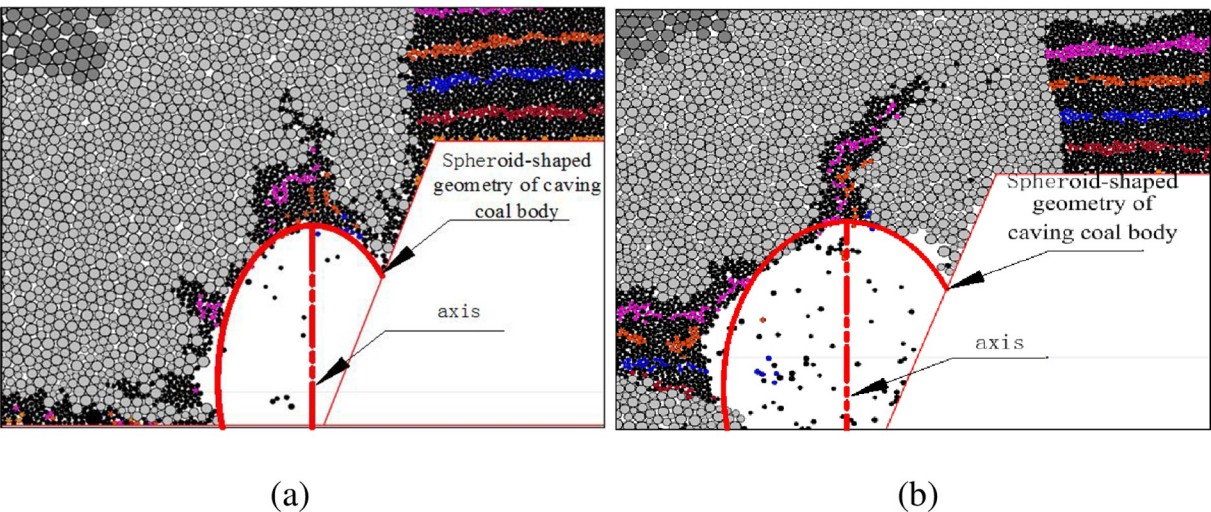

**Fig 6.** Geometry of recovered top coal: (a) below entries in upper seam; (b) below gob area in upper seam.

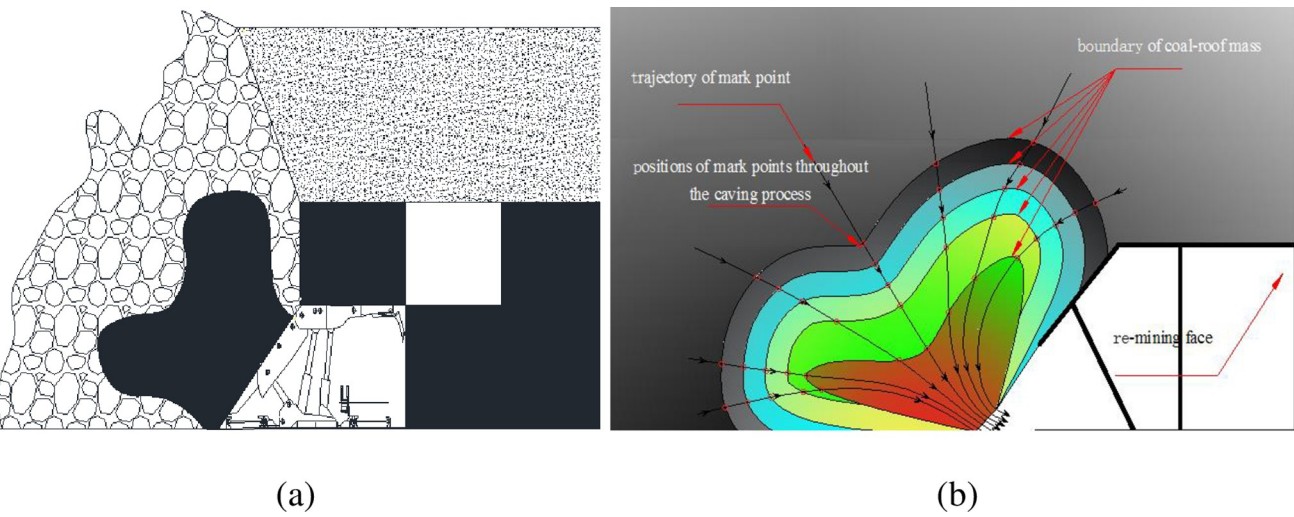

(a) (b)

**Fig 7.** (a) Funnel-shaped geometry of top coal during caving operation; (b) Progressive development of coal-roof boundary.

the waste ratio is found at 5%, at which the caving window should be closed. Caving operation may continue in order to increase top coal recovery, but further caving sees a significant increase in waste ratio but a slow growth in coal recovery. When the coal recovery reaches about 98%, the waste ratio is 100%, indicating that all the materials flowing through the caving window are rock wastes at this moment.

When the face advances to the upper entry where caving is operated, the top coal is covered with waste rocks from previously- and currently-caved immediate roof. Roof mass is found in the caved materials and recovered from the caving window at the start of caving operation. As the top coal recovery reaches 70%, the waste ratio increases to 12%, at which time cease of caving operation should be initiated. Tremendous increase of waste rocks is found if caving operation continues. As the top coal recovery is 83%, the waste ratio increases to 100%.

When the caving is operated below the upper gob area, top coal is covered with previously-caved waste rocks from the upper gob and fresh roof rocks. The beginning of caving operation sees part of the waste rocks caving through the shield window with coal mass. A 15% of waste ratio meets the condition for ceasing the caving operation, at which point the top coal recovery is about 58%. Wastes caved from shield window would increase dramatically if caving operation continues. The top coal recovery is about 72% at 100% of waste ratio.

It is seen from the above analysis that, at the beginning of wastes going through the caving window, a large amount of coal behind the shield still remains behind the shield to be recovered. The timing of caving initiation and shut down of caving window are important for maintaining a high recovery and an acceptable waste ratio. The traditional "caving operation after three cuts" may not be applicable for the re-mined face. The timing of caving operation and the proper caving interval depend on the dimensions of solid coal and configurations in upper seam.

**Table 2. Top coal recovery at different caving positions.**

| Caving position | Top coal recovered, t | Top coal lost, t | Top coal recovery |
|---|---|---|---|
| Below solid coal (4.5 m of face advance) | 0.104 | 0.002 | 98.1% |
| Below entries (9 m of face advance) | 0.027 | 0.008 | 77.1% |
| Below gob area (22.5 m of face advance) | 0.067 | 0.028 | 70.5% |

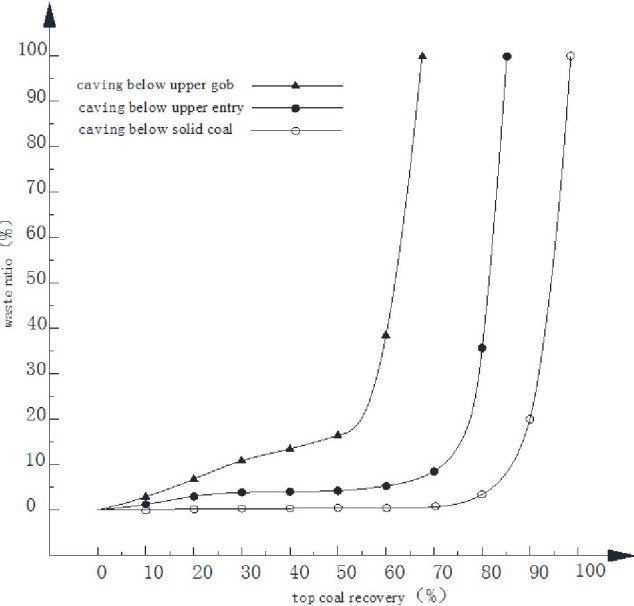

**Fig 8. Relationship between the waste ratio and coal recovery.**

## 4 Theoretical analysis of top coal flowing characteristic

The re-mined activity in the lower seam may improve the cavibility of top coal by increasing the degree of fragmentation of the upper seam and waste rocks. The caving materials above the shield window can be assumed as the viscous fluid. The caving process is simplified and analysed as the problem of unsteady flow at the thin walled nozzle. The unsteady flow model of the caving materials at the shield caving window is shown in Fig 9.

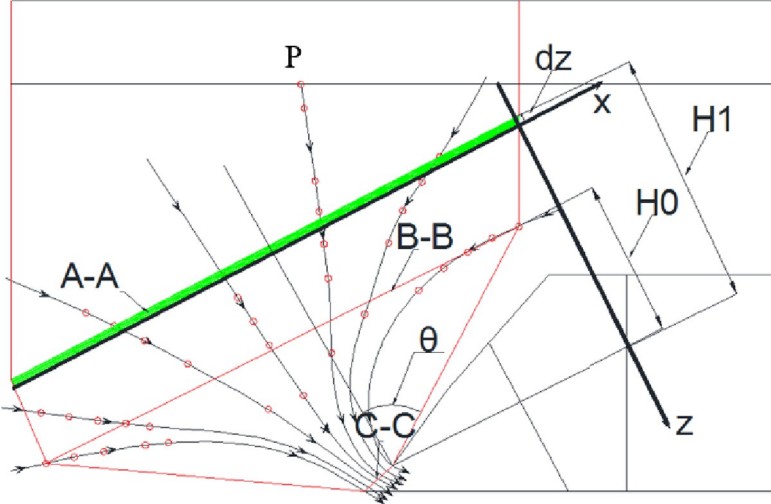

**Fig 9. Unsteady flow model of caving materials at the caving window.** Point P on the top is the vertex of the spheroid geometry in Fig 6. Section A-A is the boundary between the main roof and caved immediate roof; Section B-B is the coal-roof mass boundary before caving operation; Section C-C is the coal-roof mass boundary at the cease of caving operation. $H_1$ is the distance from caving window to Section A-A; $H_0$ is the distance from caving window to Section B-B; dz is the descending distance of Section A-A during a unit time dt.

The Bernoulli's equation between Sections A-A and B-B is given as

$$\frac{p_A}{\rho_{\text{rock}}g} + \frac{\alpha_A v_A^2}{2g} + H_{AB} = \frac{p_B}{\rho_{coal}g} + \frac{\alpha_B v_B^2}{2g} + \zeta_{AB}\frac{v_B^2}{2g} \tag{1}$$

where $p_A$ is the overlying pressure on Section A-A in MPa; $\rho_{rock}$ is density of the immediate roof in kg/m$^3$; $g$ is the acceleration of gravity given as 9.8 m/s$^2$; $\alpha_A$ is the kinetic energy correction factor selected as 2; $v_A$ is the average velocity of Section A-A; $H_{AB}$ is the shortest distance between Sections A-A and B-B in m; $p_B$ is the overlying pressure on Section B-B in MPa; $\rho_{coal}$ is density of top coal in kg/m$^3$; $\alpha_B$ is the kinetic energy correction factor taken as 2; $v_B$ is the average velocity of Section B-B; $\zeta_{AB}$ is the local resistance factor of the caving material at shield window, taken as 0.06 from tests. The equation of continuity is written as

$$v_A = \frac{A_B}{A_A}v_B \tag{2}$$

where $A_A$ is the cross-sectional area of Section A-A in m$^2$; $A_B$ is the cross-sectional area of Section B-B in m$^2$. Substituting Eq 2 into Eq 1 yields Eq 3:

$$gH_{AB} + \frac{p_A - p_B}{\rho_{coal}} = [\alpha_B - \alpha_A(\frac{A_B}{A_A})^2 + \zeta_{AB}]\frac{v_B^2}{2} \tag{3}$$

Eq 3 is re-organized for computing the average velocity of the coal-roof mass boundary (Section B-B):

$$v_B = \frac{1}{\sqrt{\alpha_B - \alpha_A\left(\frac{C_B A_B}{A_A}\right)^2 + \zeta_{AB}}}\sqrt{\frac{2\Delta p}{\rho}} \tag{4}$$

where $\triangle p = p_A - p_B \approx \gamma H_{AB}$, where $\gamma$ is the unit weight of coal. The total volume of caving materials ($q_v$) is then calculated as

$$q_v = v_B A_B \tag{5}$$

Again, the Bernoulli's equation between Sections B-B and C-C is given as

$$\frac{p_B}{\rho_{coal}g} + \frac{\alpha_B v_B^2}{2g} + H_{BC} = \frac{p_C}{\rho_{coal}g} + \frac{\alpha_C v_C^2}{2g} + \zeta_{BC}\frac{v_C^2}{2g} \tag{6}$$

where $H_{BC}$ is the shortest distance between Sections B-B and C-C in m; $p_C$ is the overlying pressure on Section C-C in MPa; $\alpha_C$ is the kinetic energy correction factor taken as 1; $v_C$ is the average velocity of Section C-C; $\zeta_{BC}$ is the local resistance factor given as

$$\zeta_{BC} = \frac{\lambda}{\sin(\theta/2)}\left[1 - (\frac{A_C}{A_B})^2\right] \tag{7}$$

where $\theta$ is the angle of widening from Section C-C to B-B in rad (see Fig 9). This angle is taken as 15–20° based on numerical modelling study. The equation of continuity is given as

$$v_B = \frac{A_C}{A_B}v_C \tag{8}$$

where $A_C$ is the cross-sectional area of Section C-C in m$^2$; Substituting Eq 8 into Eq 7 gives Eq 9:

$$gH_{BC} + \frac{p_B - p_C}{\rho_{coal}} = [\alpha_C - \alpha_B(\frac{A_C}{A_B})^2 + \zeta_{BC}]\frac{v_C^2}{2} \tag{9}$$

Re-writing Eq 9 yields the average velocity for Section C-C:

$$v_C = \frac{1}{\sqrt{\alpha_C - \alpha_B(\frac{C_C A_C}{A_B})^2 + \zeta_{BC}}}\sqrt{\frac{2\Delta p}{\rho}} \tag{10}$$

where $\triangle p = p_B - p_C \approx \gamma H_{BC}$. The total volume of caving materials ($q_v$) is then calculated as

$$q_v = v_C A_C \tag{11}$$

Assuming the coal-roof mass boundary is at a certain position on $Z$ axis with the volume of caving materials through the boundary section at $q_v$. This boundary moves downwards at a distance of $d_z$ within a period of $d_t$, then the total volume of caving materials during this period is calculated as

$$q_v dt = -A(z)dz \tag{12}$$

where $A(z)$ is instantaneous cross-sectional area of coal-roof boundary in $m^2$ as a function of $z$. The time required for top coal to drop from the height of $H_1$ to $H_0$ is given as

$$t = \int_0^t dt = -\frac{1}{q_v}\int_{H_0}^{H_1}\frac{A(z)}{\sqrt{z}}dz \tag{13}$$

where $t$ is time duration of caving operation in $s$; $H_1$ is the shortest distance from the shield caving window to the coal-roof boundary (Section A-A) at the beginning of caving operation; $H_0$ is the shortest distance from the shield caving window to the coal-roof boundary (Section B-B) at the end of caving operation. It is learned from Fig 6 that the geometry of the caving coal body is funnel-shaped. Assuming the cross-sectional area of the coal-roof boundary before caving operation is $A$, an integral operation of Eq 13 yields

$$t = \frac{2A}{q_v}(\sqrt{H_1} - \sqrt{H_0}) \tag{14}$$

Assuming the cross-sectional area of the coal-roof boundary at beginning of caving is $A_B$, and the caving operation is ceased when the coal-roof boundary reaches the caving window (or equivalently $H_0 = 0$), the time duration of caving operation is given as

$$t = \frac{2A_B}{q_v}\sqrt{H_1} \tag{15}$$

Substituting $H_1 = 2.7\ m$, $\theta = 15°$, $v_c = 4\ m/s$, $A_C = 0.09\ m^2$, $A_B = 5.27\ m^2$ into Eq 1–15 yields the time duration of caving operation of $t = 49.47\ s$.

The caving operation time calculated from Eq 15 is compared with the improved B-R model [5], which is give in Eq 16

$$r_{max} = r_D + K\left(\frac{gt^2}{2}\right)(1 - \cos\theta) \tag{16}$$

where $r_{max}$ is the displacement from Point P to the caving window (see Fig 9); $r_D$ is the width of caving window; $K$ is the correction factor for gravitational acceleration; $t$ is the time of

duration starting from the initiation of coal particle movement to its caving through the shield window. $\theta$ is selecting from 27.5~29.52˚. Substituting $r_{max} = 3\ m$, $r_D = 0.3\ m$, $K = 0.00149$, $\theta = 27.5°$ into Eq 16 yields the time of caving operation is $t = 56.63$ s.

## 5 Conclusions

1. A PFC numerical model is developed for modelling a re-mined longwall face using the top coal caving method. The movement of loose top coal and roof mass is obtained. The coal mass close to the caving window shows the largest movement velocity. Compared with the caving operation below solid coal pillar, the movement of coal-roof boundary for caving operation below upper entries and gob is relatively larger.

2. A funnel-shaped coal-roof boundary is observed above the caving window on shield, with un-recovered coal remained behind the shield in the gob. When caving operations are performed below entries and gob, a partial spheroid-shaped caving geometry of the coal body is formed. The top coal recovery is 98.1%, 77.1% and 70.5% for caving operations below solid coal, entries and gob area, respectively. A high top coal recovery at an acceptable waste ratio is expected at proper timing of caving and interval of caving operation, which depends on the geological conditions of upper seam.

3. A theoretical analysis based on unsteady flow model is developed for calculating the time of duration of caving operation, and is compared with the improved B-R model. Good agreement between two models is observed. The proposed model shows slightly less predicted time duration of caving but is more applicable for re-mined faces using top coal caving method.

## Author Contributions

**Conceptualization:** Feng He.

**Data curation:** Shuai Wang.

**Formal analysis:** Feng He, Yu Yang.

**Investigation:** Chunhua Zhang.

**Methodology:** Shuai Wang.

**Software:** Feng He, Yu Yang.

**Supervision:** Chunhua Zhang.

**Validation:** Yu Yang.

**Writing – original draft:** Shuai Wang.

**Writing – review & editing:** Chunhua Zhang.

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
