## [Decision Letter · Decision Letter 0]

13 Dec 2022

PONE-D-22-32027Numerical modelling of loose top coal and roof mass movement for a re-mined seam using the top coal caving methodPLOS ONE

Dear Dr. Wang,

Thank you for submitting your manuscript to PLOS ONE. After careful consideration, we feel that it has merit but does not fully meet PLOS ONE’s publication criteria as it currently stands. Therefore, we invite you to submit a revised version of the manuscript that addresses the points raised during the review process.

Please do your best to address all comments, specially the English language.

We look forward to receiving your revised manuscript.

Kind regards,

Khalil Abdelrazek Khalil, Ph.D.

Academic Editor

PLOS ONE

Journal Requirements:

"This research was financially supported by National Natural Science Foundation of China(2017YFC1503102), and Liaoning Natural Science Foundation(Grant No.20180550869)."

"This research was financially supported by National Natural Science Foundation of China(2017YFC1503102), and Liaoning Natural Science Foundation(Grant No.20180550869)."

"This research was financially supported by National Natural Science Foundation of China(2017YFC1503102), and Liaoning Natural Science Foundation(Grant No.20180550869)."

Additional Editor Comments:

Please do your best to address the reviewers comments.

Reviewers' comments:

Reviewer's Responses to Questions

**Comments to the Author**

1. Is the manuscript technically sound, and do the data support the conclusions?

Reviewer #1: Yes

Reviewer #2: Yes

2. Has the statistical analysis been performed appropriately and rigorously? 

Reviewer #1: Yes

Reviewer #2: Yes

3. Have the authors made all data underlying the findings in their manuscript fully available?

Reviewer #1: Yes

Reviewer #2: Yes

4. Is the manuscript presented in an intelligible fashion and written in standard English?

Reviewer #1: No

Reviewer #2: Yes

5. Review Comments to the Author

Reviewer #1: Review for manuscript: “Numerical modelling of loose top coal and roof mass movement for a re-mined seam using the top coal caving method”

This manuscript has addressed the numerical study of loose coal movement in the field of coal caving re-mined face and has developed coal-rock interface. Also an analytical formula is developed to obtain the time duration of caving operation. The manuscript idea is novel and is recommended for publication in Plos One journal subject to addressing the following comments:

1- On page 5, “Wang used the 3D physical modelling,…” Please paraphrase and improve the grammar of this sentence.

2- Page 5, line 6 “Wang used …” again this sentence needs revision.

3- On page 6 Please explain the “Mt” unit mentioned on line 10

4- On Figure 3 could you include a color bar for velocity as well?

5- Make adjustment to place Figure 8 and its caption on the same page.

6- Could you sketch the parameters in equation 16 in a separate figure?

7- Could you run your numerical model using the specified parameters in the theoretical section for the time duration and compare the outputs?

Reviewer #2: This manuscript developed a numerical model using PFC2D to study the movement of top coal mass and development of coal-rock mass interface at a longwall top coal caving re-mining face. The authors have conducted comprehensive analyses. I recommend the publication of this work after major revision.

1. Almost all the references are published by scholars in China, and some of them are even published in Chinese. I think the authors should refer to recent works of foreign scholars.

2. The numerical model is developed using PFC 2D. However, a realistic coal reservoir should be 3D. Please justify the rationality of this treatment. Why?

3. What are the red, blue, and orange lines in Figure 2? Put what they are in the caption.

4. Figure 3 also has the different colored lines. What are they representing? Put in the caption.

5.Where is the caving window in Fig 4 ,5? Show it in the figures.

Once the above concerns are fully addressed,the manuscript could be accepted for publication in this journal.

6. PLOS authors have the option to publish the peer review history of their article (what does this mean?). If published, this will include your full peer review and any attached files.

Reviewer #1: No

Reviewer #2: No

---

## [Author Response · Author response to Decision Letter 0]

17 Feb 2023

Thanks to the experts' constructive and valuable comments on the thesis, the author feels very beneficial and has revised the comments made by the experts one by one. The revised contents are as follows:

Reviewer # 1

Comment 1: On page 5, “Wang used the 3D physical modelling,…” Please paraphrase and improve the grammar of this sentence.

Reply: Changes have been made, see the section in blue on page 5.

Comment 2: Page 5, line 6 “Wang used …” again this sentence needs revision.

Reply: Changes have been made, see the section in blue on page 5.

Comment 3: On page 6 Please explain the “Mt” unit mentioned on line 10.

Reply: 5 million tons, M-million, t-ton.

Comment 4: On Figure 3 could you include a color bar for velocity as well?

Reply: Already marked in the graph, see Figure 3.

Comment 5: Make adjustment to place Figure 8 and its caption on the same page.

Reply: Already adjusted.

Comment 6: Could you sketch the parameters in equation 16 in a separate figure?

Reply: This is only a theoretical calculation, and the representation in the graph will be the next step.

Comment 7: Could you run your numerical model using the specified parameters in the theoretical section for the time duration and compare the outputs?

Reply: In this paper, numerical simulation results and B-R model (Boundary-Release model) are used to calculate the time of caving operation. The next step will be to run the numerical model for specified parameters and compare the output results.

Reviewer # 2

Comment 1: Almost all the references are published by scholars in China, and some of them are even published in Chinese. I think the authors should refer to recent works of foreign scholars.

Reply: Additions and references have been made, see page 5 of the paper.

Comment 2: The numerical model is developed using PFC 2D. However, a realistic coal reservoir should be 3D. Please justify the rationality of this treatment. Why?

Reply: Firstly, we accept the reviewer's suggestion that the actual engineering is a three-dimensional problem. According to the method of elasticity, it can be simplified to a plane strain problem, and the calculated results can be applied to three-dimensional problems.

Comment 3: What are the red, blue, and orange lines in Figure 2? Put what they are in the caption.

Reply: Simple model stratification lines with no practical meaning.

Comment 4: Figure 3 also has the different colored lines. What are they representing? Put in the caption.

Reply: Has been modified, simple model stratification line, no practical meaning.

Comment 5: Where is the caving window in Fig 4 ,5? Show it in the figures.

Reply: See Figure 4,5 (b).

---

## [Decision Letter · Decision Letter 1]

20 Mar 2023

Numerical modelling of loose top coal and roof mass movement for a re-mined seam using the top coal caving method

PONE-D-22-32027R1

Dear Dr. Wang,

We’re pleased to inform you that your manuscript has been judged scientifically suitable for publication and will be formally accepted for publication once it meets all outstanding technical requirements.

Kind regards,

Khalil Abdelrazek Khalil, Ph.D.

Academic Editor

PLOS ONE

Additional Editor Comments (optional):

Reviewers' comments:

Reviewer's Responses to Questions

**Comments to the Author**

1. If the authors have adequately addressed your comments raised in a previous round of review and you feel that this manuscript is now acceptable for publication, you may indicate that here to bypass the “Comments to the Author” section, enter your conflict of interest statement in the “Confidential to Editor” section, and submit your "Accept" recommendation.

Reviewer #1: All comments have been addressed

Reviewer #2: All comments have been addressed

2. Is the manuscript technically sound, and do the data support the conclusions?

Reviewer #1: Yes

Reviewer #2: Yes

3. Has the statistical analysis been performed appropriately and rigorously? 

Reviewer #1: Yes

Reviewer #2: Yes

4. Have the authors made all data underlying the findings in their manuscript fully available?

Reviewer #1: Yes

Reviewer #2: Yes

5. Is the manuscript presented in an intelligible fashion and written in standard English?

Reviewer #1: Yes

Reviewer #2: Yes

6. Review Comments to the Author

Reviewer #1: This grammer and structure of this sentence on page 5 could still be improved : Wang investigated the caving body geometry and interface between coal

and roof mass through the method of the 3D physical, numerical model and field

observations[12].

Reviewer #2: This manuscript developed a numerical model using PFC2D to study the movement of top coal mass and development of coal-rock mass interface at a longwall top coal caving re-mining face. The authors have adequately addressed my comments in the previous review. Comprehensive analyses have been conducted in the manuscript, and the picture in the manuscript is clear and proves the point well. The translation of the argument in the article can be understood by the reader.

7. PLOS authors have the option to publish the peer review history of their article (what does this mean?). If published, this will include your full peer review and any attached files.

Reviewer #1: No

Reviewer #2: No

---

## [Editor Report · Acceptance letter]

6 Apr 2023

PONE-D-22-32027R1 

Numerical modelling of loose top coal and roof mass movement for a re-mined seam using the top coal caving method 

Dear Dr. Wang:

I'm pleased to inform you that your manuscript has been deemed suitable for publication in PLOS ONE. Congratulations! Your manuscript is now with our production department. 

Kind regards, 

on behalf of

Dr. Khalil Abdelrazek Khalil 

Academic Editor

PLOS ONE